# Are pharmacists on the front lines of the opioid epidemic? A cross-sectional study of the practices and competencies of community and hospital pharmacists in Punjab, Pakistan

Naeem Mubarak ,[1] Taheer Zahid,[1] Fatima Rahman Rana,[1] Umm-E-Barirah Ijaz,[1] Afshan Shabbir,[1] Mahrukh Manzoor,[1] Nahan Khan,[1] Minahil Arif,[1] Muhammad Mehroz Naeem,[1] Sabba Kanwal,[1] Nasira Saif-ur-Rehman,[1] Che Suraya Zin ,[2] Khalid Mahmood,[3] Javaid Asgher,[1] Mohamed Hassan Elnaem [4]

For numbered affiliations see end of article.

**Correspondence to**
Dr Mohamed Hassan Elnaem;
m.elnaem@ulster.ac.uk

## ABSTRACT

**Introduction** Countries are grappling with a rapidly worsening upsurge in the opioid-related overdose deaths, misuse and abuse. There is a dearth of data in Pakistan regarding the practices and competencies of pharmacists in handling opioid-related issues.

**Study design** A cross-sectional study, conducted across Punjab, Pakistan.

**Method** The study deployed a validated survey to evaluate the competencies and practices of the community and hospital pharmacists.

**Results** 504 community pharmacists and 279 hospital pharmacists participated in the survey with an overall response rate of 85.5%. Almost half of the respondents 'never' or 'sometimes' made clinical notes in a journal or dispensing software to monitor ongoing opioid use. Generally, pharmacists were reluctant to collaborate with physicians or notify police regarding the abuse/misuse of opioids. Hospital pharmacists achieved significantly higher mean competency scores than chain and independent community pharmacists (p<0.05). In competency evaluation, three priority areas emerged that require additional training, that is, 'opioid overdose management', 'opioid use monitoring' and 'therapeutic uses of opioids'.

**Conclusion** Both community and hospital pharmacists hold significant positions and potential to contribute meaningfully to the mitigation of harms and risks associated with opioids. Nevertheless, this study underscores notable deficiencies in the competence of pharmacists, whether in hospital or community settings in Punjab, concerning various aspects related to the dispensing and utilisation of opioids. It also highlights the pressing need for the development of strategies aimed at improving several practice areas including the documentation, the quality of patient counselling, the effectiveness of reporting mechanisms for opioid abuse and the stringent enforcement of regulatory policies to curtail opioid misuse. Thus, to mitigate the opioid epidemic in Pakistan, it is imperative to institute opioid stewardship initiatives aimed at rectifying the competency

## STRENGTHS AND LIMITATIONS OF THIS STUDY

⇒ This study is the first of its kind to systematically assess the practices and competencies of community and hospital pharmacists in relation to opioid use and dispensing, using an extensive framework comprising eleven opioid competencies.

⇒ The study stands out for its inclusivity and generalisability, as the sample of the community and hospital pharmacists represent diversely located nine administrative divisions of the Punjab.

⇒ The length of the survey, with 55 items, could have induced respondent fatigue, potentially affecting the accuracy and completeness of the responses.

⇒ We could not comment on the absence of pharmacists in hospital settings due to the absence of data regarding the total number of pharmacists in a given hospital.

⇒ The study excluded 13 incomplete survey forms, the response rate and the diversity of the sample size achieved still support the generalisability of the results.

and procedural deficiencies within the pharmacist workforce.

## INTRODUCTION

Chronic pain is one of the most debilitating human experiences that affects over a quarter of the global population in terms of significant physical discomfort.[1] Opioid analgesics, also known as narcotics, are the primary drugs for pain relief, which strictly require a prescription of a legally qualified prescriber and notable examples include morphine, oxycodone, fentanyl, codeine and tramadol. WHO reserves opioid analgesics mainly for moderate-to-severe pain and palliative care.[2 3]



Opioids interact with their specific receptors in the brain and elsewhere to produce a powerful pain-numbing effect, a sense of invincibility and euphoria. Nevertheless, their benefits are often offset by the risks associated with their use, such as tolerance, opioid use disorder and overdose-related mortalities especially where opioids were combined with alcohol or antianxiety medicine.[4 5] Due to these inherent risks, the dispensing of opioids is subject to stringent regulations and involves meticulous documentation of the patient information, prescriber details and the exact dose in the narcotic register—a responsibility typically entrusted to a qualified pharmacist.[6 7]

'Opioid crisis' or 'opioid epidemic' refers to the alarming upsurge in opioid-related overdose deaths and opioid use disorder, primarily due to escalating opioids use/misuse. According to the 2022 World Drug Report, about 61.3 million people used opioids in the past year in 2020, in an increasing trend compared to the same statistics in 2017.[8] In 2020, approximately 70 000 opiate-related deaths were reported in the United States alone. The opioid epidemic is not confined to specific regions; it is a trend that extends globally. Illicitly manufactured fentanyl is rising in North America, Europe and Australia.[9] Asian countries are also grappling with the challenges posed by opioid misuse. Asia is the largest contributor to world's illicit opioid use, thus, the opioid epidemic remains a crucial area in global public health research because of its detrimental effect on communities, families and society, including the socioeconomic burden it imposes on healthcare systems.[10 11]

Pakistan is the fifth most populous country in the world, with its one province, Punjab, houses a population of 110 million.[12] Concerns have been mounting about the widespread misuse and abuse of prescription opioids in this colossal population. Pakistan is currently at risk, particularly due to the imported opioid epidemic from neighbouring countries. Opium production and trafficking in Afghanistan, nearly one-third of the world's opioid-using population in India and illegally manufactured opioids in China are among the major drivers.[13 14] In Pakistan, inappropriate dispensing or inept handling of opioids in community or hospital setting is considered a precipitating factor in the opioid epidemic. It has been estimated that 4.25 million adults in Pakistan use opioids unsafely and 1.6 million people reportedly misuse prescription opioids, mainly tramadol, codeine and nalbuphine.[14–16] Pharmacists, whether in community or hospital settings, play a crucial role in controlling the supply and diversion of opioids by carefully managing opioid refills, preventing individuals from purchasing opioids from multiple pharmacies, avoiding the use of irrational opioid combinations with abuse potential and monitoring the non-medical use of prescription opioids.[17 18] Further, they have a legal obligation to identify opioid misuse and diversion and collaborate with physicians and law enforcement agencies to report, manage and screen patients with opioid use disorder. Thus, pharmacists offer a comprehensive range of services to ensure rational opioid use, such as dispensing, counselling, stigma and harm reduction, overdose management, record keeping, recommending replacement therapies and safe disposal guidance.[6 19] However, efficiently delivering the above-mentioned services requires essential opioid dispensing competencies of the handling pharmacist. Opioid dispensing competencies refer to a pharmacist's knowledge, and skills to promote rational opioid use and reduce non-medical or misuse in society.[20] A recent study gained expert consensus and identified 11 key areas (competencies) for pharmacists on opioid dispensing, based on competency framework devised by the Association of Faculties of Pharmacy of Canada's Opioid Working Group.[20 21] These competencies serve as a foundation for evaluating pharmacists' professional practice standards to promote rational opioid use in each society.

Despite the severity of the opioid epidemic, there has been a dearth of studies evaluating the competence of pharmacists in Pakistan. It is important for pharmacists to ensure they are up to date with opioid dispensing guidelines. Assessing their competency in this area will provide evidence of their readiness to deal with the looming opioid epidemic and identify areas where additional training is needed, to improve their capabilities to serve global health objectives. Therefore, this study aimed to assess and compare community and hospital pharmacists' practices and competencies in dispensing, using and managing opioids in Punjab, Pakistan.

## METHODS
It was a cross-sectional study that collected data from the community and hospital pharmacists in all nine administrative divisions of the province of Punjab, in addition to Islamabad Capital Territory.

According to official sources, there were 23 029 registered pharmacists in Punjab.[22] To obtain the sampling frame for community pharmacies, we formally requested a division-wise list of all licensed community pharmacies in Punjab from the Provincial Drug Control Unit, which provided the required information. The sampling frame encompassed 10 417 community pharmacies. Similarly, the sampling frame for hospital pharmacists was obtained from the primary and secondary health official website. In Punjab, there were 664 hospital pharmacists in public sector hospitals, while when combined with pharmacists in private hospitals (approximately one-fourth of the population in public hospitals), the sampling frame reached around 830 hospital pharmacists.[23] We used 'Raosoft', an online sample size calculator, to compute an appropriate sample size for community and hospital pharmacists. Sample size calculations were based on a 5% margin of error, 95% CI, a response distribution of 50% and a maximum population of 10 417 (for community pharmacists) and 830 (for hospital pharmacists). The calculator determined a target sample size (n=371) for community pharmacists and hospital pharmacists (n=263) to represent their respective populations.

Consequently, to achieve the target sample size, we aimed at a higher number of community and hospital pharmacies, considering potential refusals and dropouts. Further, we used a stratified random sampling technique to reduce the sampling bias and obtain a diverse, representative sample of community and hospital pharmacists.

## Survey instrument

The survey instrument used the concept of 11 core opioid dispensing competencies, and survey items were informed by an extensive review of the relevant literature.[14 20 21 24–28]

The survey instrument comprised three main sections:
1. Demographic.
2. Routine practices regarding opioid use and dispensing.
3. Core opioid competencies.

The third section was further divided into 11 subsections corresponding to 11 core opioid competencies. These competencies include understanding societal, physiological and pharmacological aspects of opioids, recognising opioid therapeutic applications and managing opioid overdoses. Knowledge of pain management and adherence to state laws related to opioids was also included.[20 21] Competencies include skills like counselling, communication and understanding opioid use disorder issues, detecting vulnerable groups for signs of opioid overdose or misuse is important. Each competency was evaluated using five statements (survey items) which required the respondent to choose the correct answer out of five exclusive options. Total scores were computed using a summative scale for comparisons based on gender, pharmacy practice type, location and professional development course completion. Please refer to online supplemental appendix S1 for the full-length survey.

## Validity and reliability of the survey instrument

After an extensive literature review to develop initial survey items. Six experts, including two professors, two community pharmacists and two hospital pharmacists, evaluated the face and content validity of the items as per steps mentioned in the Yusoff *et al*.[29] They assessed each statement using two different scales: (A) relevance, with options (1) 'not relevant', (2) 'somewhat relevant', (3) 'quite relevant' or (4) 'highly relevant', and (B) clarity, with options (1) 'not clear', (2) 'somewhat clear', (3) 'quite clear' or (4) 'highly clear'. The item-level Content Validity Index (I-CVI) for relevance was calculated as I-CVI (R)=Nr/N, where Nr represents the number of experts who rated an item as (3) or (4) and N is the total number of experts involved. Statements with an I-CVI in the range of 0.83 to 1.00 were deemed acceptable, while those below 0.83 were rejected. This process resulted in a reduction from the initial 70 statements (items) in the questionnaire to a final version comprising 55 items divided into 11 domains related to opioids competency. Moreover, to ensure the internal consistency/ reliability of the survey items, a pilot study involving 50 pharmacists (33 community pharmacists and 17 hospital pharmacists) was conducted, and Cronbach's alpha statistics were applied. The results of the reliability analysis showed that the items in the 11 competencies had a satisfactory level (0.7–0.9) of internal consistency ('see online supplemental table S1').

## Definitions, inclusion and exclusion criteria

In the context of this study, a pharmacy within the premises of a hospital was referred to as a hospital pharmacy, which can further be categorised based on the type of hospital they are associated with, such as public or private hospitals. On the other hand, any pharmacy outlet outside the premises of a hospital was defined as a community pharmacy. While the existing scholarly work lacks a consensus-based definition of a chain or an independent pharmacy. Chain pharmacies are typically a corporation with 6–200+ pharmacies operating under a nationwide banner. On the other hand, an independent pharmacy follows a small operator model with an individual owning up to five pharmacies. Thus, for our research, we defined a pharmacy as a chain if more than five pharmacies were operating under the same name and the business used distinctive branding across all pharmacies. Detailed inclusion/exclusion criteria are given in table 1.

| Table 1 | Inclusion–exclusion criteria |
| --- | --- |
| **Inclusion** | **Exclusion** |
| Any community or hospital pharmacy registered under a category A licence, located in any of the nine divisions of Punjab or Islamabad Capital Territory, was eligible to be included in the study. | Pharmacies without a category A licence or medical stores with category B licence, or designated dispensaries were excluded from data collection. |
| Pharmacists registered in category A (legally permitted to handle opioids) were included. | Assistant pharmacist or pharmacy technicians or pharmacy interns (who had not completed the Doctor of Pharmacy degree) were excluded. |
| Secondary or tertiary care hospitals both in the public and private sector were included. | Primary care clinics/facilities were excluded because generally the facility is smaller and do not have a pharmacist within the premises especially in the private sector. |
| Survey forms completed in all respects were included in the final count. | Incomplete/missing survey forms were excluded. |

## Data collection and reporting

The first author conducted a comprehensive training of a team of Doctor of Pharmacy (PharmD) students and research associates on different aspects of data collection. The team was divided into subgroups assigned to different divisions of Punjab. Data collectors were instructed to pay a second visit to a community pharmacy/a hospital pharmacy, if the pharmacist was unavailable, or the pharmacy was occupied in rush hours during the first visit. The survey was administered in a face-to-face meeting. Data collection began on 10 December 2022 and ended on 15 January 2023. The data were reported in accordance with Strengthening the Reporting of Observational Studies in Epidemiology guidelines (see online supplemental appendix S2).

## Data management and statistical analysis

We used Statistical Package Social Sciences (V.22) for data management and statistical analysis. Descriptive statistics were used to evaluate categorical variables, while differences in mean scores were compared between two groups for different variables using a t-test. To conduct quantitative analysis, we made groups based on different variables, such as gender, nature of the employment and type of pharmacy practice. To determine the mean difference among three or more groups, we conducted one-way analysis of variance (ANOVA) followed by post hoc analysis. The significance level for detecting differences in means was set at a p value of ≤0.05.

## Patient and public involvement

No patient was involved in this survey.

## RESULTS

Data were collected from community and hospital pharmacies diversely located in all administrative divisions of the province of Punjab, namely Lahore (provincial capital), Sargodha, Multan, Faisalabad, Bahawalpur, Rawalpindi, Sahiwal and Dera Ghazi Khan besides Islamabad Capital Territory. During the first visit, pharmacists were present in only 44.8% of the pharmacies visited. This presence of pharmacists was further declined when the research team moved from the provincial capital to other cities, and the lowest presence (18.9%) was recorded in Faisalabad. However, the presence increased significantly on the second visit. A total of (n=783) pharmacists completed the survey (community pharmacist=504; hospital pharmacist=279). Most respondents were full-time employed men within the age range of 26–30 years. A mere 22.6% of the pharmacists reported involvement in any continuous professional development programme within the past 2 years. Table 2 provides a summary of the demographics of the survey participants.

The overall response rate collectively for community and hospital pharmacists was 85.5%, as shown in online supplemental table S2. However, it is essential to note that many pharmacists (mostly in hospitals) declined to participate in the survey due to reasons, such as a busy schedule, lack of motivation or fear of potential regulatory consequences.

The survey also examined the trends of professional practices on opioid use and dispensing among community and hospital pharmacists. Half of the respondents mentioned that they 'never' or 'sometimes' made clinical notes in a journal or dispensing software to monitor ongoing opioid use. Roughly one-third of the respondents reported that they 'almost always' counsel patients on the possible side effects and overdose risks associated with opioid use. Similarly, only 18% reported that they 'almost always' contacted police or physicians regarding the abuse of opioids or clarify an opioid prescription. Table 3 presents the trends of practice among pharmacists.

Statistical analysis revealed no significant difference in the overall mean competency scores against the variables, such as gender, community versus hospital settings or system of assessment used in the PharmD degree programme or type of hospital (private or public). In contrast, full-time employed pharmacists with a higher degree after PharmD, as well as those who had completed continuous professional development courses, achieved significantly higher mean competency scores. Similarly, pharmacists with a degree from a public university scored slightly higher (significance at borderline) than pharmacists who received a PharmD degree from a private educational institute. Table 4 summarises the overall mean competency score against each variable.

On the other hand, results of ANOVA revealed that statistically significant differences (p=0.006) exist among the three pharmacist groups compared, that is, hospital, chain and independent pharmacist. Hospital pharmacists secured higher mean competency scores than all other subgroups (both chain and independent). Further, the mean competency scores for pharmacists at chain pharmacies were better than pharmacists at independent pharmacies, as shown in table 5.

Online supplemental table S3 details the mean score for each of the eleven individual opioid competencies against different variables. Comparatively, among the 11 opioid competencies, community pharmacists received the lowest mean competency score in 'therapeutic use of opioids' (C-5=1.34), 'opioid and society' (C-1=1.39) and 'opioid use monitoring' (C-10=1.41) thus emerged as a priority area to focus for further training. For hospital pharmacists, 'opioid use monitoring' (C-10=1.45), 'therapeutic use of opioids' (C-5=1.52), 'opioid and society' (C-1=1.56) and 'opioid overdose management' (C-11=1.56) had the lowest mean scores.

## DISCUSSION

The findings of this study portray a bleak picture of the current professional practices and competencies of pharmacist workforce related to opioid use and dispensing in Punjab, Pakistan.

**Table 2** Demographics of the survey participants

| Variable | Category | | n (%) | |
|---|---|---|---|---|
| Gender | Male | | 503 (64.2) | |
| | Female | | 280 (35.8) | |
| Age group | 23–25 | | 190 (24.3) | |
| | 26–30 | | 367 (46.9) | |
| | 31–35 | | 139 (17.8) | |
| | 36–40 | | 57 (7.3) | |
| | 41–45 | | 19 (2.4) | |
| | 46–50 | | 9 (1.1) | |
| | 56–60 | | 2 (0.3) | |
| | Total | | 783 (100) | |
| You studied/completed PharmD/B. Pharm degree in a ________ institute. | Public | | 403 (51.5) | |
| | Private | | 380 (48.5) | |
| Mode of assessment in your PharmD/B. Pharm was: | Annual | | 366 (46.7) | |
| | Semester | | 417 (53.3) | |
| Your highest level of education | Masters | | 155 (19.8) | |
| | PharmD/B. Pharm | | 628 (80.2) | |
| Have you completed any continuous professional development course in the last two years? | Yes | | 177 (22.6) | |
| | No | | 606 (77.4) | |
| Employment status | Full time | | 614 (78.4) | |
| | Part-time | | 169 (21.6) | |
| Type of pharmacy practice | Community | Chain | 291 (37.2) | 504 (64.4) |
| | | Independent | 213 (27.2) | |
| | Hospital | Public | 187 (67) | 279 (35.6) |
| | | Private | 92 (33) | |

PharmD, Doctor of Pharmacy.

First, it is pertinent to mention that pharmacist availability, especially in the community setting, was challenging during the first visit to pharmacies. Our study confirms the earlier findings of Bashir et al, which reported the absence of pharmacists in 50% of the pharmacies in different regions of Punjab.[30] Legally, only a registered pharmacist can dispense opioids in Pakistan. It is not some kind of delegated responsibility, but an inherent role of pharmacist as defined through an act of law.[13] Pharmacies without a qualified pharmacist or operating on a proxy could precipitate an upsurge in the dispensing of opioids without a prescription and pervasive misuse

**Table 3** Practices among pharmacists on opioid use

| Statements | Never n (%) | Rarely n (%) | Sometimes n (%) | Very often n (%) | Almost always n (%) |
|---|---|---|---|---|---|
| I______ make clinical notes in a journal/dispensing software (apart from the narcotic register) to monitor ongoing opioids use. | 179 (22.9) | 166 (21.2) | 178 (22.7) | 121 (15.5) | 139 (17.8) |
| I ______ counsel patients on the possible side effects of opioids. | 26 (3.3) | 89 (11.4) | 187 (23.9) | 217 (27.7) | 264 (33.7) |
| I ______ counsel patients on opioids overdose risk. | 40 (5.1) | 86 (11.0) | 184 (23.5) | 189 (24.1) | 284 (36.3) |
| I ______ refuse the supply of the opioids when in doubt of abuse. | 36 (4.6) | 75 (9.6) | 105 (13.4) | 161 (20.6) | 406 (51.9) |
| I ______ notify police/regulatory bodies on the abuse of opioids. | 203 (25.9) | 154 (19.7) | 145 (18.5) | 135 (17.2) | 146 (18.6) |
| To clarify the opioids prescription, I ______ call the physician. | 76 (9.7) | 140 (17.9) | 294 (37.5) | 129 (16.5) | 144 (18.4) |

**Table 4** Different variables against the summative mean score of all the core opioids competencies

| Variable | Category | Mean (SD) | T* (p value) |
|---|---|---|---|
| Gender | Male | 19.64 (8.301) | −0.816 (0.415) |
| | Female | 20.14 (8.019) | |
| System of assessment in your PharmD degree programme | Annual | 20.25 (8.455) | 1.393 (0.164) |
| | Semester | 19.44 (7.959) | |
| Type of institute | Public | 20.36 (8.700) | 1.921 (0.055)* |
| | Private | 19.24 (7.602) | |
| Highest level of education | Masters | 21.99 (9.099) | **3.408 (0.001)*** |
| | PharmD/B. Pharm | 19.28 (7.878) | |
| Employment status | Full time | 20.50 (8.180) | **4.519 (0.000)*** |
| | Part-time | 17.33 (7.800) | |
| Have you completed any continuous professional development course in the last two years? | Yes | 20.86 (9.101) | **2.079 (0.038)*** |
| | No | 19.41 (7.787) | |
| Type of pharmacy practice | Community | 19.40 (7.869) | −1.846 (0.065) |
| | Hospital | 20.57 (8.730) | |
| If pharmacy is in hospital, type of hospital: | Public | 21.05 (9.047) | 1.316 (0.189) |
| | Private | 19.59 (8.004) | |

Bold values indicate statistical significance.
*Independent t-test, p value (≥0.05).
PharmD, Doctor of Pharmacy.

and abuse of prescription opioids. Further, this finding implies poor regulatory check and balance and raises serious concerns on the consequences of low engagement of community pharmacists in opioid management. Thus, pharmacists, especially in community settings, are not visible at the front line. The government should ensure the implementation of the mandatory presence of a qualified person (pharmacist) in community pharmacies in Punjab. The loopholes in the current regulatory policies and the partial implementation of the Control of Narcotic Substances Act 1967 are also attributed to opioid misuse and unauthorised dispensing. A few countries, including India, Ukraine and Colombia, have recently amended their laws to engage community pharmacists to offer opioid stewardship activities, though the implementation is slow.[10 31]

**Table 5** Comparison of different types of pharmacy practice

| Type of pharmacy practice | Groups | Mean difference | P value |
|---|---|---|---|
| Chain | Independent | 1.904 | 0.010* |
| | Hospital | −0.357 | 0.602 |
| Independent | Chain | −1.904 | 0.010* |
| | Hospital | −2.261 | 0.002* |
| Hospital | Chain | 0.357 | 0.602 |
| | Independent | 2.261 | 0.002* |

*P value (≤0.05).

Second, findings of this study reveal a potential gap in the documentation practices, which are crucial for ensuring patient safety and continuity of care. Nevertheless, studies support the exigent need for community pharmacist engagement to facilitate the implementation of safe opioid use in society.[6 18] This situation is further exacerbated by the low engagement of pharmacists in professional development programmes, which emphasises the need for initiatives to make continuous education and training programmes a regulatory prerequisite for practice. Currently, continuing education for pharmacists is not mandatory in Pakistan. Our findings strongly suggest that making continuing education mandatory for pharmacists could lead to an overall improvement in the quality of healthcare delivery, particularly in the context of opioid dispensation. Another recent research also endorsed that prior training results in better competency of community pharmacists in handling medication opioid use disorder.[15] Thus, community and hospital pharmacists must undergo continuous professional education to dispense opioids and follow standard guidelines.

Third, unfortunately, only one-third of the pharmacists reported that they 'almost always' counselled patients on the possible side effects and overdose risks associated with opioid use. While pharmacists have the professional scope to educate patients about opioid-related risks and benefits, suggest alternatives, conduct medication reviews as part of an interdisciplinary team and counsel on opioid substitution therapy and management of opioid poisoning.[6 32] These findings may be correlated with the recent surge in the abuse of opioid analgesics, particularly

Nalbuphine, in Pakistan.[15] Another study in Pakistan also attributed unchecked dispensing at pharmacies with much of the opioid's abuse and misuse in Pakistan. Thus, to curtail the opioid-related overdose mortalities, enhanced patient education especially in the community setting could improve the opioid use in the society.

Fourth, pharmacists were reluctant to call the physician to clarify opioid-related prescriptions, indicating a lack of interprofessional collaboration between pharmacists and general practitioners. These findings indicate the importance and urgent need to enhance interprofessional communication and collaboration, especially in community settings. The decision-making process for a pharmacist in determining whether to contact the police or the prescriber in response to a suspicious or potentially problematic opioid prescription is a multifaceted one. Contacting the police will be the last reserve for illegal suspicious orders that have not proven to be authentic on validation with the prescriber. It is not an option in every case to contact the police versus prescriber. Thus, there are no rigid, one-size-fits-all guidelines for when to contact the police versus the prescriber regarding an opioid prescription; however, there are general principles and practices that guide pharmacists in making this critical decision. It is important to note that these principles are contextual and may vary depending on local regulations, institutional policies and individual judgement.[13 14] The safety and well-being of the patient will always be the paramount consideration. Pharmacists generally evaluate the prescription's legitimacy and appropriateness for the patient's medical condition. If there is a reasonable suspicion that the prescription may pose a risk to the patient's health or well-being, contacting the prescriber to seek clarification is usually the first step. Pharmacists are trained to recognise signs of forged or fraudulent prescriptions. If there are indications that the prescription is not legitimate or has been altered, it is advisable to contact the prescriber to validate its authenticity. Pharmacists are encouraged to assess the prescribing history of a patient, especially if they notice a recurring pattern of early refills, high doses or multiple prescriptions from different prescribers. Such patterns may raise red flags and necessitate further investigation or contact with the prescriber.[13 14] An Australian study identified that authentication of the prescription by contacting the prescriber as one of the crucial strategies used in prescription opioid monitoring.[17] Ensuring appropriate opioid use requires devising a mechanism or channel of communication between the prescriber and pharmacist, especially in a community setting in Pakistan. Numerous studies provide evidence for the positive outcomes of collaborative care on medication reviews, identifying prescribing errors and intervening in opioid misuse in other countries.[33 34] Similarly, only a handful of respondents reported that they notified the police or regulatory bodies regarding the abuse of opioids, highlighting a potential lack of awareness or reluctance to report such cases. The collaboration of pharmacists with law enforcement agencies, opioid use disorder consultants, policy-makers, regulators, epidemiologists and pharmaceutical companies is equally important.

Fifth, higher competency scores of hospital pharmacists compared with chain and independent pharmacies may be linked to their higher exposure and clinical and patient care experience in hospital settings. Similarly, better competency scores at chain pharmacies compared with their counterparts in independent pharmacies could be attributed to the fact that chain pharmacies hold more financial resources, which they spent on the training and development of their pharmacists.

Finally, the pharmacist who had studied under the current PharmD curriculum in Pakistan lack sufficient competencies to practice across all scopes of opioid management and dispensing. The lowest overall mean competency scores in 'opioid overdose management', 'opioid use monitoring' and 'therapeutic use of opioids' indicate the top three priority areas which require further training and education. These findings align with the conclusion drawn in a recent qualitative study on the dearth of skill-based competency among pharmacists, such as identifying and communicating prescription errors, therapeutic duplication, monitoring unauthorised dispensing and providing patient-centred care.[15] A Canadian study also validated similar key areas of training in opioid and pain management, alternative pain management, naloxone dispensing, collaborating with physicians and counselling patients on opioid use and misuse.[35] Another possible reason for these poor competency scores may be related to the limited coverage of these topics in the PharmD curriculum in Pakistan. Further, the PharmD curriculum in Pakistan has not been revised for a decade, making it difficult for fresh graduates to adapt to the updated guidelines on pain management and opioid use.[36] This highlights the urgent need to upgrade the PharmD curriculum and incorporate opioid learning modules with updated guidelines. Lessons should be learnt from the case of the USA, where, in line with the recent opioid epidemic, pharmacy schools are incorporating specific opioid-related competencies into their PharmD curriculums to ensure that students are well prepared to handle the complex issues surrounding opioid use and misuse.[37 38]

### Implications for policy, practice and future research

Based on the findings of this study, the authors suggest the following implications for policy, practice and further research:

1. It is crucial to address the gaps in knowledge and practice related to therapeutic use of opioids, opioid use monitoring and opioid overdose management. Improving pharmacist competency in these areas could contribute to the safe and effective use of opioids, thereby reducing the risks associated with their misuse and abuse.
2. Strategies should be developed to promote the documentation of clinical notes and enhanced patient counselling on opioid risks.

3. The notification of opioid misuse to regulatory authorities and clarification of opioid-related prescription from the physicians should be made a binding requirement.

4. To curb opioid misuse, the availability of a qualified personnel at community pharmacist should be ensured.

5. Community and hospital pharmacists must pass a licensing exam that assesses the different opioid dispensing competencies of pharmacists before recruitment in community and hospital settings.

6. An electronic national opioid database system should be developed to store the records of all opioid prescriptions. This will assist in tracing the unauthorised prescribing and dispensing of opioids.

7. The PharmD curriculum should be revised, and opioids must be covered in depth at the graduate and undergraduate level primarily related to opioid use disorder, misuse and diversion. A detailed and comprehensive module of opioid use can be integrated in pharmacology, clinical pharmacy and forensic pharmacy. The module's components can be decided on various consensus methods, such as Delphi and nominal group technique, etc.

8. Future research should prioritise the examining of policies and interventions aimed at ensuring the active participation of pharmacists in community pharmacies. Furthermore, there is a need for research to focus on the development of evidence-based opioid stewardship interventions, clinical support tools and collaborative practice models fostering enhanced communication among the pharmacist, prescribers and regulatory bodies.

The main strengths of this work are related to being the pioneer study to report the practices and competencies of community and hospital pharmacists on opioid use and dispensing in a systematic way that is, under the framework of eleven opioid competencies. It thus may be taken as a call for action. Our study did not select a convenient sample of the community pharmacies and hospital pharmacists, instead, it reached out to all nine diverse divisions of Punjab to present data representative of the population.

The current study should be interpreted in the light of its limitations. It was a long survey consisting of 55 items that may have resulted in the fatigue bias of the respondent. However, it was inevitable in the context of the coverage of essential 11 core competencies related to opioid use and dispensing. In the case of community pharmacy, the presence percentage was possible because one pharmacist/pharmacy generally follows the law of the land. However, we could not comment on the absence of pharmacists in hospital settings due to the absence of data regarding the total number of pharmacists in a hospital. We did not include 13 incomplete survey forms; however, given the response rate, diverse and vast sample size achieved, the findings remain the representative of the population and results are expected to be generalisable.

## Conclusion

While the opioid epidemic continues to inflict grave and unsettling consequences, both community and hospital pharmacists hold significant positions and potential to contribute meaningfully to the mitigation of harms and risks associated with opioids at both individual and societal levels. Nevertheless, this study underscores notable deficiencies in the competence and practices of pharmacists, whether in hospital or community settings in Punjab, concerning various aspects related to the dispensing and utilisation of opioids. It also highlights the pressing need for the development of strategies aimed at improving several practice areas including the documentation, the quality of patient counselling, the effectiveness of reporting mechanisms for opioid abuse and the stringent enforcement of regulatory policies to curtail opioid misuse. Thus, to combat the looming opioid epidemic in Pakistan, it is crucial to equip pharmacists with the necessary competencies to offer opioid stewardship activities to ensure responsible opioid use. However, achieving this goal necessitates a forward-thinking approach, an eagerness to collaborate and notify physicians and regulatory authorities, a willingness to honestly introspect and acknowledge the lack of competencies or past failed efforts and an investment in the training of pharmacy human resources. Embracing this life-saving commitment is imperative for the well-being of the patients, communities and the pharmacy profession.

**Author affiliations**
[1]Lahore Medical and Dental College, Lahore University of Biological and Applied Sciences, Lahore, Pakistan
[2]Department of Pharmacy Practice, Kulliyyah of Pharmacy, International Islamic University Malaysia, Kuantan, Malaysia
[3]University of the Punjab Quaid-i-Azam Campus, Lahore, Pakistan
[4]School of Pharmacy and Pharmaceutical Sciences, Ulster University, Coleraine, UK

**Contributors** NM: Conceptualisation, methodology, formal analysis, writing—original draft preparation, writing—review and editing, resources, supervision. TZ, U-E-BI, AS, NK, MA and MMN: Methodology, investigation, resources, writing—review and editing. FRR: Methodology, investigation, formal analysis, writing—original draft preparation, writing—review and editing, resources. MM: Methodology, investigation, resources. SK: Methodology, formal analysis, resources, writing—review and editing. NS-u-R: Methodology, investigation, resources, supervision, writing—review and editing. CSZ and JA: Methodology, investigation, writing—review and editing, resources, supervision. KM and MHE: Methodology, formal analysis, writing—review and editing, resources, supervision. NM is responsible for the overall content as the guarantor.

**Funding** The authors have not declared a specific grant for this research from any funding agency in the public, commercial or not-for-profit sectors.

**Competing interests** None declared.

**Patient and public involvement** Patients and/or the public were not involved in the design, or conduct, or reporting, or dissemination plans of this research.

**Patient consent for publication** Not applicable.

**Ethics approval** This study involves human participants and was approved by the research ethics committee, Lahore Pharmacy College, Lahore Medical & Dental College granted ethical approval for this study (ref# LPC/ETH/25/09/22) (see online supplemental material S3 appendix). Participants gave informed consent to participate in the study before taking part.

**Provenance and peer review** Not commissioned; externally peer reviewed.

**Data availability statement** All data relevant to the study are included in the article or uploaded as online supplemental information.

**ORCID iDs**
Naeem Mubarak http://orcid.org/0000-0002-0547-2168
Che Suraya Zin http://orcid.org/0000-0002-5572-2719
Mohamed Hassan Elnaem http://orcid.org/0000-0003-0873-6541

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
