## [Reviewer comments · BMJ Open]

ARTICLE DETAILS

TITLE (PROVISIONAL)	"Are Pharmacists on the Frontlines of the Opioid Epidemic? A Cross Sectional Study of the Practices and Competencies of Community and Hospital Pharmacists in Punjab, Pakistan"
AUTHORS	MUBARAK, NAEEM; Zahid, Taheer; Rana, Fatima Rahman; Ijaz, Umm-E-Barirah; Shabbir, Afshan; Manzoor, Mahrukh; Khan, Nahan; Arif, Minahil; Naeem, Muhammad Mehroz; Kanwal, Sabba; Saif-ur-Rehman, Nasira; Zin, Che; Mahmood, Khalid; Asgher, Javaid; Elnaem, Mohamed Hassan

VERSION 1 – REVIEW

REVIEWER	Hayat, Khezar Xi'an Jiaotong University School of Medicine, department of Pharmacy Administration and Clinical Pharmacy
REVIEW RETURNED	22-Sep-2023

GENERAL COMMENTS	I have carefully reviewed the manuscript titled "Are Pharmacists at the Frontline? A Cross Sectional Analysis of the Practices and Competencies of Community and Hospital Pharmacists Amid the Looming Opioid Crises." The study aimed to evaluate the competencies and practices of pharmacists in handling opioid-related issues in Punjab, Pakistan. Below are my critical comments and suggestions for improvement:  1. Clarity and Structure: The manuscript is well-structured, with clear objectives, methods, results, and conclusions. However, the introduction could benefit from a more detailed discussion regarding the global opioid crisis and its impact on Pakistan, emphasizing the urgency and significance of the study. 2. Sample Size and Representativeness: While the response rate of 85.5% is commendable, providing information about the total number of hospital and community pharmacists approached and their demographics would enhance the understanding of the representativeness of the sample. Similarly, the contradiction of the response rate of 85.5% and the conclusion pharmacists were absent in majority do not make sense and needs to be justified. 3. Methodology and Survey Validity: The methodology section though provides some details on the validity and reliability, however, more detailed information about the survey tool's development, validation, and reliability would enhance the clarity on these two important issues. Providing this information would ensure the robustness and validity of the study's findings. 4. Results and Analysis: The results are presented clearly, including the Mean Competency Scores and priority areas for additional training. However, subgroup analysis requires a bit more details and discussion of the differences in competencies between hospital and community pharmacists would enhance the manuscript's rigor.
--

	5. Discussion: This is the essence of this manuscript and is well written and presented. 6. Conclusion: The conclusion effectively summarizes the study's key findings and emphasizes the need for improving pharmacist competencies and practices to combat the opioid crisis. Overall, the study addresses an important issue which is relevant not only for Pakistan but for other countries in the region. It would be like a wake-up call and serves the purpose from an introspection point of view. Enhancements in the areas mentioned above would significantly strengthen the manuscript's quality and impact. I recommend accepting the review with minor revisions.
--	--

REVIEWER	Khan, Barkat Gomal University Faculty of Pharmacy
REVIEW RETURNED	30-Sep-2023

GENERAL COMMENTS	Thank you for an interesting read. The manuscript presents a comprehensive study on the competencies and practices of hospital and community pharmacists in handling opioids in the context of the opioid crisis in Pakistan. While the manuscript addresses a pressing public health concern, opioid misuse and overdose, which is of significant relevance not only in Pakistan but also globally. However, there are several strengths and areas for improvement that should be considered before publication. Strengths: 1. Relevance and Significance: The manuscript addresses an important and timely issue – the opioid crisis – and its impact on Pakistan. Given the global nature of this crisis, the findings of this study are relevant not only to Pakistan but also to other countries facing similar challenges. 2. Methodological Rigor: The study employs a cross-sectional design with a well-defined sampling strategy, enhancing the robustness of the findings. The use of validated competencies and a comprehensive survey instrument adds to the methodological strength of the study. 3. Data Collection and Reporting: The study's extensive data collection across different regions of Punjab and Islamabad is commendable. The reporting adheres to STROBE guidelines, ensuring transparency and completeness in reporting. The inclusion of both community and hospital pharmacists provides a comprehensive view of the issue. 4. Identification of Priority Areas: The identification of priority areas for additional training, such as "opioid overdose management," "opioid use monitoring," and "therapeutic use of opioids," is a valuable contribution to the field. These findings can inform targeted educational programs for pharmacists. Areas for Improvement:
--

	1. Literature Review and Theoretical Framework: While the introduction provides an overview of the opioid crisis and its significance in Pakistan, it lacks a comprehensive literature review and theoretical framework to contextualize the study within existing research and theory. Incorporating a more extensive review of relevant literature and theoretical concepts would strengthen the manuscript. 2. Validation of Survey Instrument: The manuscript mentions the validation of the survey instrument but lacks specific details about the validation process, including how experts assessed face and content validity. Providing more information on this aspect would enhance the credibility of the survey. 3. Data Presentation and Analysis: The presentation of results in tables is clear and informative. However, the discussion of results in the text could be more concise and focused. Rather than summarizing each competency separately, the discussion should emphasize key findings, implications, and their alignment with the study's objectives. 4. Interprofessional Collaboration: The manuscript highlights the importance of interprofessional collaboration but does not delve into the barriers and facilitators of such collaboration among pharmacists and other healthcare professionals. Including insights into these factors would enrich the discussion. 5. Pharmacy Curriculum: The manuscript rightly identifies the need for revising the Pharm-D curriculum in Pakistan to include updated guidelines on pain management and opioid use. However, it would be beneficial to provide specific recommendations or strategies for curriculum reform in the conclusion. 6. Ethical Approval: The manuscript mentions obtaining ethical approval, which is essential. However, it would be helpful to clarify whether informed consent was obtained from survey participants and how their anonymity was ensured. In conclusion, this manuscript addresses an important issue and provides valuable insights into the competencies and practices of pharmacists in the context of the opioid crisis in Pakistan. To enhance its quality and impact, the authors should consider addressing the above-mentioned areas for improvement. Once these aspects are refined, the manuscript has the potential to make a significant contribution to the field of pharmacy practice and public health. Thus, I recommend acceptance with minor revisions.
--	--

REVIEWER	Tutag Lehr, Victoria Wayne State University
REVIEW RETURNED	02-Oct-2023

GENERAL COMMENTS

Thank you for allowing me to review your manuscript. This study addresses an important research question yet there are several areas requiring clarification/revision:

1. Title: The title is very long and does not accurately describe the study-Consider eliminating “Are pharmacists on the frontline?” Focus on the survey aims which appear to be “knowledge and competencies of pharmacists regarding opioid prescriptions”
2. Key words require revising-several do not pertain to the key points of the survey study –appears that the Key words on the title page differ from those provided with the abstract –need to consolidate/clarify which are correct? Please use MESH headings
3. Abstract requires organization into the customary categories of Intro/Background, Objectives, Methods, Results, Discussion/Conclusion. Please check journal requirements Does not reflect key points of the survey study -
4. Introduction: The introduction would be improved by eliminating the first paragraph on chronic pain and role of opioid analgesics and replacing with a “snapshot” of the opioid epidemic landscape in Pakistan and the contribution of prescription opioids to opioid overdose deaths (using current data). Recommend a brief summary of the expectations of the pharmacist when dispensing prescription opioids, including monitoring and any role in overdose prevention education. Have the requirements for pharmacist practice changed since the opioid epidemic? Use preferred term epidemic vs “crisis”. Have pharmacy programs in India updated their curricula to include any training on opioid overdose and misuse? How about training on collaboration and starting the conversation with prescribers (physicians)? Who else can dispense opioids in a community pharmacy? Is this a delegated responsibility adding to inappropriate or unsafe opioid dispensing or use?
When does a pharmacist determine to contact the police vs the prescriber regarding an opioid prescription? Are there guidelines? There appear to be multiple factors contributing here...Have dispensing guidelines for pharmacists in Pakistan been updated? Do pharmacists have mandatory continuing education that includes opioids? Is there a difference in training based on practice site (community vs hospital practice?)
Briefly describe the above for an international audience despite detailing in discussion in relation to results.
Line 99 page 4: “addiction” replace with Opioid Use Disorder – please use non stigmatizing language throughout the manuscript <https://nida.nih.gov/nidamed-medical-health-professionals/health-professions-education/words-matter-terms-to-use-avoid-when-talking-about-addiction>
5. Cross-sectional design? This is a survey study designed to determine the knowledge and self-assessed competencies of pharmacists regarding opioid prescriptions. May not be able to extrapolate the knowledge and competencies to opioid misuse, overdose events and overdose deaths in the population. If there is a difference in hospital vs community pharmacist practice in dispensing prescriptions to the public, it may be best to focus on reporting community pharmacists results for this paper (unless hospital pharmacists have a role in educating physicians in prescribing opioids in the hospital setting)
6. Survey Instrument: Line 135 page 6: please use the term survey here. In reviewing some of the survey items, have found some of the wording to be confusing with ambiguous responses, however may reflect the translation to English. In addition, would expect more application based items to assess pharmacist

	competency. Was surprised at inclusion of the CDC guidelines based items (expect this was the 2016 version vs 2022 revision-please clarify?). 7. Survey methods: How where the research surveyors trained for consistent face to face interviews? Was there introductory material that they read to the prospective participant? What level of program where the PharmD students? How many surveyors were involved in the study? Were survey responses de-identified? 8. Statistical analysis: Please explain calculation of the competency scores for the various categories. Consider use of regression to identify pharmacist and practice related factors associated with specific outcome (eg competency score). Reconsider use of ANOVA for your analysis-suggest consult a statistician. 9. Results: recommend reporting results in the text section by subheadings of survey: refer to tables as needed, do not repeat information from tables. 9. References- Please recheck ref #17 regarding Pharmacist Competencies-please cite original competency source, this is a survey of Australian pharmacist competency Ref#17: This cites a multidisciplinary statement, not specific to pharmacy- did you mean this Opioid Summit statement vy Canadian CPHA? https://www.pharmacists.ca/cpha-ca/assets/File/cpha-on-the-issues/Pharmacy%20Opioid%20Summit.pdf 10. Limitations: In addition to those provided, others include responder bias, face to face survey (participants may not be anonymous), lack of external validity, low response rate, some participants may have interpreted selected survey items differently, primarily male, less 30 years of age participating in survey (does this reflect the local pharmacist workforce?) and self assessed competencies. 11. Conclusion: Unclear regarding point #5, line #55, page 16- why only community pharmacists? Unclear regarding this point. Points #4 and #8 are stating the same concept regarding the responsibility of a pharmacist in a community pharmacy in a community pharmacy to dispense opioids. These points can be summarized. What is the action plan? Next steps? Communication of results? Understand that English is a difficult language, however your manuscript would benefit from review by a native English speaker for clarity and succinctness.
--	--

VERSION 1 – AUTHOR RESPONSE

REVIEWER-1

Clarity and Structure: The manuscript is well-structured, with clear objectives, methods, results, and conclusions. However, the introduction could benefit from a more detailed discussion regarding the global opioid crisis and its impact on Pakistan, emphasizing the urgency and significance of the study.

We have added a few more lines to satisfy the worthy reviewer’s valid comment, however, a major limitation is the word count posed by the journal. (Line # 103-113, Page # 4-5)

Sample Size and Representativeness: While the response rate of 85.5% is commendable, providing information about the total number of hospital and community pharmacists approached and their demographics would enhance the understanding of the representativeness of the sample. Similarly, the contradiction of the response rate of 85.5% and the conclusion pharmacists were absent in majority do not make sense and needs to be justified.

Table-2 provides a holistic picture of the participating pharmacists' demographics in community and hospital settings. (Line # 243, Page # 10)

The response rate of 85.5% was mentioned by calculating the number of pharmacists who finally completed the survey (included both on first and second visit). On the other hand, the absence from the pharmacies were computed on the first visit.

Methodology and Survey Validity: The methodology section though provides some details on the validity and reliability, however, more detailed information about the survey tool's development, validation, and reliability would enhance the clarity on these two important issues. Providing this information would ensure the robustness and validity of the study's findings.

More details on the process of validation have been provided. (Line # 169-183, Page#7)

Results and Analysis: The results are presented clearly, including the Mean Competency Scores and priority areas for additional training. However, subgroup analysis requires a bit more details and discussion of the differences in competencies between hospital and community pharmacists would enhance the manuscript's rigor.

More details have been provided. (Line # 282-288, Page # 13)

Discussion: This is the essence of this manuscript and is well written and presented. Thank you for the positive feedback on this section.

Conclusion: The conclusion effectively summarizes the study's key findings and emphasizes the need for improving pharmacist competencies and practices to combat the opioid crisis.

Thank you so much.

Overall, the study addresses an important issue which is relevant not only for Pakistan but for other countries in the region. It would be like a wake-up call and serves the purpose from an introspection point of view. Enhancements in the areas mentioned

above would significantly strengthen the manuscript's quality and impact. I recommend accepting the review with minor revisions.

Many thanks for this feedback acknowledging the manuscript's quality and contribution to the existing literature.

REVIEWER-2

Relevance and Significance: The manuscript addresses an important and timely issue – the opioid crisis – and its impact on Pakistan. Given the global nature of this crisis, the findings of this study are relevant not only to Pakistan but also to other countries facing similar challenges.

Thank you so much.

Methodological Rigor: The study employs a cross-sectional design with a well-defined sampling strategy, enhancing the robustness of the findings. The use of validated competencies and a comprehensive survey instrument adds to the methodological strength of the study.

Thank you so much.

Data Collection and Reporting: The study's extensive data collection across different regions of Punjab and Islamabad is commendable. The reporting adheres to STROBE guidelines, ensuring transparency and completeness in reporting. The inclusion of both community and hospital pharmacists provides a comprehensive view of the issue.

Thank you so much.

Identification of Priority Areas: The identification of priority areas for additional training, such as "opioid overdose management," "opioid use monitoring," and "therapeutic use of opioids," is a valuable contribution to the field. These findings can inform targeted educational programs for pharmacists.

Thank you so much.

Areas for Improvement: Literature Review and Theoretical Framework: While the introduction provides an overview of the opioid crisis and its significance in Pakistan, it lacks a comprehensive literature review and theoretical framework to contextualize the study within existing research and theory. Incorporating a more extensive review of relevant literature and theoretical concepts would strengthen the manuscript.

We have updated the introduction and discussion part in line with the suggestion of the worthy reviewer. (Line # 103-113, Page # 4-5)

Validation of Survey Instrument: The manuscript mentions the validation of the survey instrument but lacks specific details about the validation process, including how experts assessed face and content validity. Providing more information on this aspect would enhance the credibility of the survey.

More details on the process of validation have been provided. (Line # 169-183, Page # 7)

Data Presentation and Analysis: The presentation of results in tables is clear and informative. However, the discussion of results in the text could be more concise and focused. Rather than summarizing each competency separately, the discussion should emphasize key findings, implications, and their alignment with the study's objectives.

The discussion section is updated in line with the suggestion. (Line # 295-306,330-337, Page # 13-15)

Interprofessional Collaboration: The manuscript highlights the importance of interprofessional collaboration but does not delve into the barriers and facilitators of such collaboration among pharmacists and other healthcare professionals. Including insights into these factors would enrich the discussion.

We would choose to humbly disagree with the worthy reviewer. Highlighting barriers and facilitators for interprofessional collaboration is beyond the scope of this study and would lead the manuscript away from the original research questions and objectives. Further, the data collected during research do not highlight the underlying factors but depict a snapshot of the current practices.

Pharmacy Curriculum: The manuscript rightly identifies the need for revising the Pharm-D curriculum in Pakistan to include updated guidelines on pain management and opioid use. However, it would be beneficial to provide specific recommendations or strategies for curriculum reform in the conclusion.

Thank you for this valuable suggestion. We have updated this in the recommendation sections. (Line # 371-376, Page # 16)

Ethical Approval: The manuscript mentions obtaining ethical approval, which is essential. However, it would be helpful to clarify whether informed consent was obtained from survey participants and how their anonymity was ensured.

Informed consent (written) was obtained, and anonymity of the response was ensured by assigning a fictitious number to survey respondents.

In conclusion, this manuscript addresses an important issue and provides valuable insights into the competencies and practices of pharmacists in the context of the opioid crisis in Pakistan. To enhance its quality and impact, the authors should consider

addressing the above-mentioned areas for improvement. Once these aspects are refined, the manuscript has the potential to make a significant contribution to the field of pharmacy practice and public health. Thus, I recommend acceptance with minor revisions.

Many thanks for this feedback acknowledging the manuscript's quality and contribution to the existing literature.

REVIEWER-3

We appreciate your valuable feedback and would like to address the specific points you raised in your review.

In completing this review, I received assistance from my trainees:

Khola Ghulzar PharmD candidate WSU 2024 and Mohamed Ayoub PharmD candidate WSU 2024

Thank you for allowing me to review your manuscript. This study addresses an important research question yet there are several areas requiring clarification/revision:

Title: The title is very long and does not accurately describe the study. Consider eliminating "Are pharmacists on the frontline?" Focus on the survey aims which appear to be "knowledge and competencies of pharmacists regarding opioid prescriptions".

We acknowledge your comment regarding the title of our manuscript. In response to your suggestion, we have indeed updated the title. However, it is important to clarify that the primary focus of our study was not solely on "knowledge and competencies." Instead, the emphasis was placed on the evaluation of "competencies" and "practices" among pharmacists. We hope this adjustment better reflects the core objectives of our research.

Second, we would like to clarify the rationale behind the line "Are pharmacists at the frontline?" in our manuscript. This statement reflects a major finding of our study, which revealed a notable absence of pharmacists during the initial visits to community pharmacies. This observation is a critical aspect of our research, as it underscores a potential gap in the role of pharmacists in the frontline healthcare system. We believe that highlighting this finding is essential for understanding the context and implications of our study.

Key words require revising-several do not pertain to the key points of the survey study – appears that the Key words on the title page differ from those provided with the abstract – need to consolidate/clarify which are correct? Please use MESH headings.

We have updated the key words in line with your suggestion. Thank you so much. (Line

66, Page # 2)

Abstract requires organization into the customary categories of Intro/Background, Objectives, Methods, Results, Discussion/Conclusion. Please check journal requirements. Does not reflect key points of the survey study.

Thank you so much. We have updated the abstract in line with your suggestion. (Line # 36, Page # 2)

Introduction: The introduction would be improved by eliminating the first paragraph on chronic pain and role of opioid analgesics and replacing with a “snapshot” of the opioid epidemic landscape in Pakistan and the contribution of prescription opioids to opioid overdose deaths (using current data). Recommend a brief summary of the expectations of the pharmacist when dispensing prescription opioids, including monitoring and any role in overdose prevention education.

We have introduced a snapshot of the opioid epidemic landscape in Pakistan, incorporating the most current available data to illustrate the gravity of the situation. We also emphasize the contribution of prescription opioids to opioid overdose deaths, thereby highlighting the immediate public health concern. However, we respectfully disagree with the suggestion to eliminate the paragraph on pain management and the role of opioid analgesics. We believe that this paragraph is essential for the following reasons:

Opioids are primarily employed in the management of moderate to severe pain, which aligns with the World Health Organization (WHO) guidelines on pain management. Including this paragraph in our introduction is crucial for providing context and significance to our study.

The paragraph on pain management is directly relevant to one of the competency areas of pharmacists explored in our study. It is imperative to maintain this paragraph to ensure that readers can connect the content of our research to the expected roles and responsibilities of pharmacists in pain management.

Have the requirements for pharmacist practice changed since the opioid epidemic? Use preferred term epidemic vs “crisis”.

Thank you so much. We have updated the introduction part in line with your suggestion. (Line # 81, Page # 4)

Have pharmacy programs in India updated their curricula to include any training on opioid overdose and misuse?

What we mentioned was the policy not the curriculum. (Line # 304-306, Page # 14)

How about training on collaboration and starting the conversation with prescribers (physicians)? Who else can dispense opioids in a community pharmacy?

Legally, only a registered pharmacist can dispense opioids in Pakistan. Regarding the interaction with the prescriber, we have addressed this query in the subsequent response. (Line # 295-297, Page # 13)

Is this a delegated responsibility adding to inappropriate or unsafe opioid dispensing or use?

It is not a delegated responsibility. It is the inherent role of pharmacist as defined through an act of law, however, our study highlighted multiple malpractices among the pharmacists.

When does a pharmacist determine to contact the police vs the prescriber regarding an opioid prescription? Are there guidelines?

At the time of interaction with the patient during dispensation process. The decision-making process for a pharmacist in determining whether to contact the police or the prescriber in response to a suspicious or potentially problematic opioid prescription is a multifaceted one. Contacting the police will be the last reserve for illegal suspicious orders that have not proven to be authentic upon validation with the prescriber. It is not an option in every case to contact the police vs prescriber Thus, there are no rigid, one-size-fits-all guidelines for when to contact the police versus the prescriber regarding an opioid prescription, however, there are general principles and practices that guide pharmacists in making this critical decision. It is important to note that these principles are contextual and may vary depending on local regulations, institutional policies, and individual judgment.

The safety and well-being of the patient will always be the paramount consideration.

Pharmacists generally evaluate the prescription's legitimacy and appropriateness for the patient's medical condition. If there is a reasonable suspicion that the prescription may pose a risk to the patient's health or well-being, contacting the prescriber to seek clarification is usually the first step.

Pharmacists are trained to recognize signs of forged or fraudulent prescriptions. If there are indications that the prescription is not legitimate or has been altered, it is advisable to contact the prescriber to validate its authenticity.

Pharmacists are encouraged to assess the prescribing history of a patient, especially if they notice a recurring pattern of early refills, high doses, or multiple prescriptions from different prescribers. Such patterns may raise red flags and necessitate further investigation or contact with the prescriber.

There appear to be multiple factors contributing here...Have dispensing guidelines for pharmacists in Pakistan been updated?

Yes, Drug Regulatory Authority of Pakistan issued updated dispensing guidelines for high alert medication management (opioids included) in 2022.

Do pharmacists have mandatory continuing education that includes opioids?

Currently, continuing education for pharmacists is not mandatory in Pakistan. Our findings strongly suggest that making continuing education mandatory for pharmacists could lead to an overall improvement in the quality of healthcare delivery, particularly in the context of opioid dispensation.

Is there a difference in training based on practice site (community vs hospital practice?)

In response to your question, we would like to provide a more comprehensive and

structured answer to highlight the distinctions in training between community and hospital pharmacists:

Hospital pharmacist's role typically involves direct interaction with healthcare providers and patients, making them an integral part of the healthcare team. Training programs for hospital pharmacists often encompass in-depth pharmacotherapy knowledge, advanced clinical assessments, and experience in medication management for complex medical conditions.

In contrast, community pharmacists place a greater emphasis on the business aspects of pharmacy practice. They have lesser interaction with physicians. Further, community pharmacists tend to have a higher volume of prescription dispensing and OTC inquiries with opportunities for extended patient counselling. Their primary role often centres around managing the dispensing of medications, maintaining inventory, and providing essential counselling to patients regarding optimal use of the medications.

Finally, the practice environment also significantly differs between community and hospital settings. Hospital pharmacists work in a structured, often interprofessional, healthcare environment with access to patient medical records and collaboration with healthcare teams. Community pharmacists, on the other

hand, work in retail settings and must manage a variety of tasks, including customer service, inventory management, and prescription processing.

Briefly describe the above for an international audience despite detailing in discussion in relation to result.

We have incorporated these points in the discussion for international audience. Thank you for this suggestion. (Line # 295-297,330-337, Page # 13-15)

Line 99 page 4: "addiction" replace with Opioid Use Disorder –please use non stigmatizing language throughout the manuscript <https://nida.nih.gov/nidamed-medical-health-professionals/healthprofessions-education/words-matter-terms-to-use-avoid-when-talking-about-addiction>

We have reviewed the whole manuscript and replaced these words with more appropriate expression. Many thanks.

Cross-sectional design? This is a survey study designed to determine the knowledge and self-assessed competencies of pharmacists regarding opioid prescriptions. May not be able to extrapolate the knowledge and competencies to opioid misuse, overdose events and overdose deaths in the population.

Our study is indeed based on a cross-sectional design, as you correctly noted. However, it is crucial to emphasize that our primary aim was to assess the "competencies" and "practices" of pharmacists concerning opioid use, dispensing, and overdose management, with an additional focus on their competency in identifying cases at risk of opioid misuse. It is not solely centred on knowledge or competency aspects, as implied. In fact, our study extensively explores current practices related to opioid dispensing among pharmacists. Therefore, it is essential to underscore that the data collected, and the subsequent analysis naturally lend themselves to the examination of opioid misuse, overdose, and overdose-related deaths within the population.

By evaluating the competencies of pharmacists, we gain insight into their preparedness to handle opioid prescriptions, including identifying patients at risk of misuse. This information can directly impact the prevention of opioid misuse and related problems.

The study assesses the current practices of pharmacists, which encompasses the actual behaviours and routines in opioid dispensing. Understanding these practices is essential to appreciating the real-world impact of pharmacists on opioid-related outcomes. Thus, the data collected during our cross-sectional study will provide insights into the opioid dispensing practices of pharmacists,

potentially shedding light on areas that may contribute to misuse and overdose. This information is valuable for designing targeted interventions to reduce misuse and its associated risks.

If there is a difference in hospital vs community pharmacist practice in dispensing prescriptions to the public, it may be best to focus on reporting community pharmacists results for this paper (unless hospital pharmacists have a role in educating physicians in prescribing opioids in the hospital setting)

Thanks for your suggestion, however, hospital pharmacists have a major role in educating physicians. Furthermore, addition of hospital pharmacist permits a comparison of competencies between the two groups strategically located at two different positions. This enabled us to conduct a comprehensive assessment of the pharmacy workforce that could play a role in mitigating this crisis.

Survey Instrument: Line 135 page 6: please use the term survey here. Thank you. We have updated it to word survey.

In reviewing some of the survey items, have found some of the wording to be confusing with ambiguous responses, however, may reflect the translation to English. In addition, would expect more application-based items to assess pharmacist competency. Was surprised at inclusion of the CDC guidelines-based items (expect this was the 2016 version vs 2022 revision-please clarify?).

The inclusion of CDC guidelines-based items in our study aimed to assess the extent to which pharmacists are informed about current best practices in overdose management. Most of the survey items were designed to evaluate clinical applications in overdose management, thereby serving as an essential aspect of our research.

To provide further clarity, we utilized the most recent CDC guidelines available during the preparation of our survey, which was the 2022 revised version. The intention behind this choice was to ensure that the survey questions align with the most current and evidence-based international guidelines. In the fast-evolving landscape of healthcare, it is imperative that pharmacists, are knowledgeable about the latest guidelines to provide optimal care and reduce harm.

Survey methods: How where the research surveyors trained for consistent face to face interviews?

For face-to-face survey administration, students were trained prior to data collection in a conference room on different aspect of opioids and any possible query. They

were also provided with the project information sheet as well as ethical approval of the study from the university ethical review board.

Was there introductory material that they read to the prospective participant?

Yes. A project information sheet was provided to the surveyors to share with the participants before survey administration.

What level of program were the PharmD students? They were Pharm D final year students.

How many surveyors were involved in the study?

15 surveyors were sent to different remote regions in the province of Punjab and Islamabad Capital Territory.

Were survey responses de-identified?

We assigned fictitious codes for each respondent and keep the privacy of the respondent intact.

Statistical analysis: Please explain calculation of the competency scores for the various categories. Consider use of regression to identify pharmacist and practice related factors associated with specific outcome (e.g., competency score). Reconsider use of ANOVA for your analysis-suggest consult a statistician.

We have carefully considered your comments, especially the one concerning the statistical analysis and the choice of statistical methods.

First, it is important to note that our statistical analysis was conducted in collaboration with Prof. Dr. Khalid Mahmood, a statistician who is also a co-author of this manuscript. Dr. Mahmood possesses extensive expertise in the field of statistics and was actively involved in the selection of appropriate statistical methods.

While we understand your suggestion to consider regression analysis for identifying factors associated with specific outcomes, we believe that the use of analysis of variance (ANOVA) aligns well with the objectives and design of our study. ANOVA is typically employed when comparing three or more groups to evaluate differences in continuous data. In our study, we had multiple groups and sought to assess variations in competency scores across these groups. ANOVA was chosen as an appropriate statistical method to achieve this objective.

In our specific context, the primary aim was to determine whether there were significant differences in competency scores across different categories and groups. The use of regression, while a valuable technique for assessing associations and predicting outcomes, might not be the most suitable method for addressing this research question. Regression is often employed when the focus is on modelling the relationships between independent variables and dependent variables. In our study, we were primarily concerned with group-level comparisons and the influence of factors on overall competency scores, rather than building a predictive model.

Nonetheless, we would like to reassure you that the decision to use ANOVA was made after careful consideration of the research objectives and the nature of our data.

Results: recommend reporting results in the text section by subheadings of survey: refer to tables as needed, do not repeat information from tables.

Thanks, we considered this comment in the revised version. However, the word limit restricted to some extent, the addition of further text and extend the subheadings.

References- Please recheck ref #17 regarding Pharmacist Competencies-please cite original competency source, this is a survey of Australian pharmacist competency
Ref#17: This cites a multidisciplinary statement, not specific to pharmacy- did you mean this Opioid

Summit statement Canadian CPHA?

<https://www.pharmacists.ca/cphaca/assets/File/cphatheissues/Pharmacy%20Opioid%20Summit.pdf>

The conceptual framework underpinning our survey was developed with a strong foundation in the eleven "consensus-based" competencies tailored to pharmacists, as established in the Delphi study referenced in source 17. It is important to note that this Delphi study do encompassed items derived from the Canadian summit, (ref 18). Thus, we cited both concurrently in our work. Consequently, our survey methodology is firmly rooted in a comprehensive understanding of the profession's core competencies, as established through rigorous research and expert consensus.

Limitations: In addition to those provided, others include responder bias, face to face survey (participants may not be anonymous), lack of external validity, low response rate, some participants may have interpreted selected survey items differently, primarily male, less 30 years of age participating in survey (does this reflect the local pharmacist workforce?) and self-assessed competencies.

We appreciate your insightful feedback and thoughtful considerations regarding the limitations of our study. In response to the limitations you raised, we offer the following clarifications and justifications:

In our study, the hard copy of the questionnaire was indeed delivered to participants in person. We opted for this mode of delivery to differentiate it from surveys sent via mail or online links. As you correctly noted, traditionally, self-administered surveys have been distributed either by mail or in-person to a large group of people and completed using paper and pencil. In our case, we followed a similar approach by providing participants with a hard copy of the survey instrument.

The choice of in-person delivery was guided by several considerations. Firstly, our research design aimed to maintain the traditional self-administered nature of surveys, as defined by respondents independently completing the questionnaire without assistance from the researcher. By distributing the surveys in person, we ensured to provide answer or clarification to any query on the survey items on spot. Secondly, in person delivery also ensures a higher response rate compared with mailed survey.

It is essential to clarify that our participants did not have any prior knowledge of the surveyors or other participants, in line with the standard protocol for maintaining the anonymity and impartiality of the survey process. We took rigorous measures to maintain the confidentiality of responses and protect the participants' privacy. The survey was designed to ensure anonymity, with participants not knowing the surveyors or other participants. This approach aimed to encourage candid responses and mitigate responder bias to the best extent possible.

It is worth noting that our study achieved a response rate of 85%, which is considered quite high for self-administered surveys. Given this high response, diverse and random sample, the findings may be generalized in Pakistan.

The potential variation in how participants interpret survey items is a generalized concern in survey research and not unique to our study. Further, we made efforts to mitigate this by ensuring that survey questions were clear and unambiguous (a trained, qualified data collector was physically present to explain any query on the survey items or its meaning).

The demographic composition of our sample, primarily consisting of participants under the age of 30, is indeed reflective of the local pharmacist workforce. This was confirmed through extensive preliminary research, and the inclusion of a

diverse sample from various administrative divisions supports the representativeness of our findings.

It is essential to clarify that the competencies in our survey were not self-assessed. Each survey item was designed to have a correct answer, and responses were graded by our research team to ensure objectivity and reliability.

In summary, we believe our study has been conducted with a rigorous methodology that addresses many of these concerns.

Conclusion: Unclear regarding point #5, line #55, page 16-why only community pharmacists? Unclear regarding this point. Points #4 and #8 are stating the same concept regarding the responsibility of a pharmacist in a community pharmacy in a community pharmacy to dispense opioids. These points can be summarized. What is the action plan? Next steps? Communication of results?

We acknowledge your comment with respect to point# 5 and have added hospital pharmacists to clear the confusion. Point 4 guides on the mandatory presence of pharmacists in community pharmacists while point 8 refers to future research areas for how policy development can help ensure maximum presence of pharmacists in community settings.

We aim to communicate the findings to Drug Regulatory Authority of Pakistan as well as to the Provincial Drug Control Unit.

VERSION 2 – REVIEW

REVIEWER	Hayat, Khezar Xi'an Jiaotong University School of Medicine, department of Pharmacy Administration and Clinical Pharmacy
REVIEW RETURNED	31-Oct-2023
GENERAL COMMENTS	The authors have addressed most of my concerns. There are no further comments.
REVIEWER	Khan, Barkat Gomal University Faculty of Pharmacy
REVIEW RETURNED	27-Oct-2023
GENERAL COMMENTS	I read the revised version of the article and I am satisfied with replies to my queries.